**Data Availability Statement:** There are no primary data in the paper, all experimental data was

# Interneuronal network model of theta-nested fast oscillations predicts differential effects of heterogeneity, gap junctions and short term depression for hyperpolarizing versus shunting inhibition

**Guillem Via**[1⊙], **Roman Baravalle**[1⊙], **Fernando R. Fernandez**[2], **John A. White**[2], **Carmen C. Canavier**[1]*

**1** Louisiana State University Health Sciences Center, Department of Cell Biology and Anatomy, New Orleans, Louisiana, United States of America, **2** Department of Biomedical Engineering, Center for Systems Neuroscience, Neurophotonics Center, Boston University, Boston, Massachusetts, United States of America

⊙ These authors contributed equally to this work.

* ccanav@lsuhsc.edu

## Abstract

Theta and gamma oscillations in the hippocampus have been hypothesized to play a role in the encoding and retrieval of memories. Recently, it was shown that an intrinsic fast gamma mechanism in medial entorhinal cortex can be recruited by optogenetic stimulation at theta frequencies, which can persist with fast excitatory synaptic transmission blocked, suggesting a contribution of interneuronal network gamma (ING). We calibrated the passive and active properties of a 100-neuron model network to capture the range of passive properties and frequency/current relationships of experimentally recorded PV+ neurons in the medial entorhinal cortex (mEC). The strength and probabilities of chemical and electrical synapses were also calibrated using paired recordings, as were the kinetics and short-term depression (STD) of the chemical synapses. Gap junctions that contribute a noticeable fraction of the input resistance were required for synchrony with hyperpolarizing inhibition; these networks exhibited theta-nested high frequency oscillations similar to the putative ING observed experimentally in the optogenetically-driven PV-ChR2 mice. With STD included in the model, the network desynchronized at frequencies above ~200 Hz, so for sufficiently strong drive, fast oscillations were only observed before the peak of the theta. Because hyperpolarizing synapses provide a synchronizing drive that contributes to robustness in the presence of heterogeneity, synchronization decreases as the hyperpolarizing inhibition becomes weaker. In contrast, networks with shunting inhibition required non-physiological levels of gap junctions to synchronize using conduction delays within the measured range.

## Author summary

Fast oscillations nested within slower oscillations have been hypothesized to play a role in the encoding and retrieval of memories by chunking information within each fast cycle;

previously published. The simulations in the paper can be reproduced using the code publicly available at https://senselab.med.yale.edu/modeldb/enterCode?model=267338#tabs-1.

**Funding:** This work was funded by NIH NS054281 to CCC and JAW. GV received full salary support and RB, FRF, JAW and CCC received partial salary support from NIH NS054281. The funders had no role in study design, data collection and analysis, decision to publish, or preparation of the manuscript.

**Competing interests:** The authors have declared that no competing interests exist.

networks of parvalbumin positive inhibitory interneurons contribute to the generation of fast oscillations. We show that, in the entorhinal cortex, the intrinsic dynamical properties of these neurons are sufficiently heterogeneous that electrical synapses are likely required to synchronize fast oscillations. Moreover, synchrony likely requires the chemical synapses to have a reversal potential that is negative relative to the action potential threshold of individual neurons during these oscillations. We show that the range of slow phases that support a fast oscillation is controlled by short term synaptic depression. The precise phase locking of the fast oscillation within the slow oscillations is hypothesized to allow for multiplexing of information.

## Introduction

Theta and gamma oscillations in the hippocampus have been hypothesized to play a role in the encoding and retrieval of memories [1–5]. Recent evidence supports the hypothesis that decrements in the temporal precision with which gamma power is coupled to a specific theta phase underlie the decline of associative memory in normal cognitive aging in humans [6]. The medial entorhinal cortex (mEC) generates fast gamma that is thought to convey information about current sensory information to other hippocampal areas [7]. Parvalbumin positive (PV+) neurons are known to contribute to gamma generation; however, the mechanism by which they contribute may differ depending upon the circumstances [8, 9]. In pyramidal interneuronal network gamma (PING) models [10], reciprocal coupling between pyramidal cells and inhibitory interneurons is required to sustain the oscillation, whereas in interneuronal network gamma (ING) models [11], only reciprocal connectivity between inhibitory neurons is sufficient to sustain the oscillation. Recently, it was shown that an intrinsic fast gamma mechanism in mEC can be recruited by optogenetic stimulation at theta frequencies in transgenic mice expressing ChR2 under the Thy1 promoter; Thy1 is expressed in both excitatory and inhibitory neurons [12]. In that study, blocking excitatory transmission abolished theta nested gamma synchrony. However, a more recent study by Butler et al., 2018 [13] in transgenic mice expressing ChR2 under a CaMKIIα promoter found that gamma oscillations were decreased in amplitude but still prominent when excitatory synaptic transmission was blocked. We have also observed fast oscillations nested within optogenetic theta in PV ChR2 mice [14], with presumably little to no contribution from excitatory synapses. Thus, it seems that the contribution of interneuronal interactions to fast oscillations generated in mEC may be variable. In this study, we have attempted to faithfully capture heterogeneity in the intrinsic and synaptic properties of PV+ fast spiking basket cells in a specific region, the medial entorhinal cortex (mEC). We previously performed dual intracellular patch recordings from layer 2/3 mEC in male and female mice expressing the tdTomato fluorophore in PV+ cells [15], and further analyzed that data in this study. We examine the effect of heterogeneity, gap junctions, synaptic depression, and the synaptic inhibitory reversal potential on synchronization of fast oscillations nested within an excitatory theta drive.

The current study differs from previous studies [11,16,17] on robustness of fast oscillations in inhibitory interneuronal networks to heterogeneity in two principal aspects: 1) the excitability type of the interneuron models and 2) the way in which intrinsic heterogeneity is introduced into the interneuronal network. First, there are two main dynamical mechanisms by which repetitive spiking can arise, corresponding to an early classification of excitability types 1 and 2 [18]. Neurons with type 1 excitability can fire repetitively at arbitrarily slow rates, act as integrators [19], with spiking arising from a saddle node bifurcation [20]. Neurons with type 2 excitability cannot fire repetitively below an abrupt cutoff frequency, act as resonators [19], and their spiking generally arises from a subcritical Hopf bifurcation [20]. Our recent

work [21] shows that PV+ fast spiking interneurons in medial entorhinal cortex neurons likely exhibit type 2 excitability, which is consistent with measures also indicating type 2 excitability in striatum [22] and neocortex [23]. Therefore, the model we constructed of the PV+ neurons has type 2 excitability. Second, previous studies used the bias current as the source of heterogeneity but kept the intrinsic dynamics constant. In contrast, in our study, the passive and active parameters of the model were sufficiently variable across the 100 neurons in the network to capture the full range in the experimentally observed f/I curves. The distinction in the implementation of heterogeneity, along with possible regional differences between mEC and other brain areas, led us to find that, in contrast to the previous modeling results, hyperpolarizing rather than shunting inhibition confers more robustness to heterogeneity.

## Results

### Model calibration

Our objective was to capture the heterogeneity in the interneuronal population as faithfully as possible. Whereas many previous studies kept the f/I curve for the neurons constant and simply varied rheobase by manipulating the bias current, we emphasized fitting the envelopes of the f/I curves across the population. Step currents were applied to determine whether physiological neurons (Fig 1A1) or model neurons (Fig 1A2) could support repetitive firing at various levels of current as shown in Fig 1B. The stabilized steady frequency was recorded (see Methods) and plotted as the f/I curves (Fig 1B). While maintaining the approximate distribution of cutoff frequencies (the minimum frequencies below which repetitive firing could not be sustained) and rheobases, we also attempted to simultaneously fit multiple additional constraints, described below, that ensured the voltage traces observed during simulated measurement of the f/I curves (Fig 1A2) were similar to the experimentally observed ones (Fig 1A1) using only values of passive properties in the measured physiological range. The resting potential, time constant and input resistance ranges are consistent with an earlier study [24] and given in the Methods. The action potential (AP) amplitude (~40 mV above threshold), half-width (~0.3 ms) and after-hyperpolarizion (AHP) depth (~20 mV) were constrained within the range of experimentally observed values. The parameters for the Na and Kv3 currents were calibrated (see Methods) in the absence of Kv1 to reproduce the action potential waveform. Moreover, in some model neurons accumulation of Na inactivation and Kv1 activation were calibrated to exhibit the weak early spike frequency adaptation observed in some recorded neurons, with the additional detail that in some model neurons $K_V1$ parameters were calibrated to emit one or more spikes then fall silent at values of injected current too weak to sustain repetitive firing (Fig 1A). Our results are consistent with previous studies showing that Kv1 and Kv3 set the minimum and maximum firing rates, respectively [25, 26]. Applying all of these constraints to the selection of model neurons did not allow us to honor the shape of the distributions of all measured passive properties, rheobase and cutoff frequency exactly (S1A–S1E Fig); however, all model values are within the experimentally observed ranges. The intrinsic parameters were frozen at the values that generated Fig 1B2, and only connectivity parameters were varied in the subsequent simulations. Fig 2 shows the experimental histograms and fits for the distribution of peak conductances in chemical and electrical synapses, with parameters and connection probabilities given in the Methods section.

### Response of homogeneous networks with hyperpolarizing versus shunting synapses to theta modulation: phase response curve analysis

After calibrating the model network, we assessed its response to a theta-modulated external input that simulates an optogenetic protocol to study its synchronizing properties. To gain

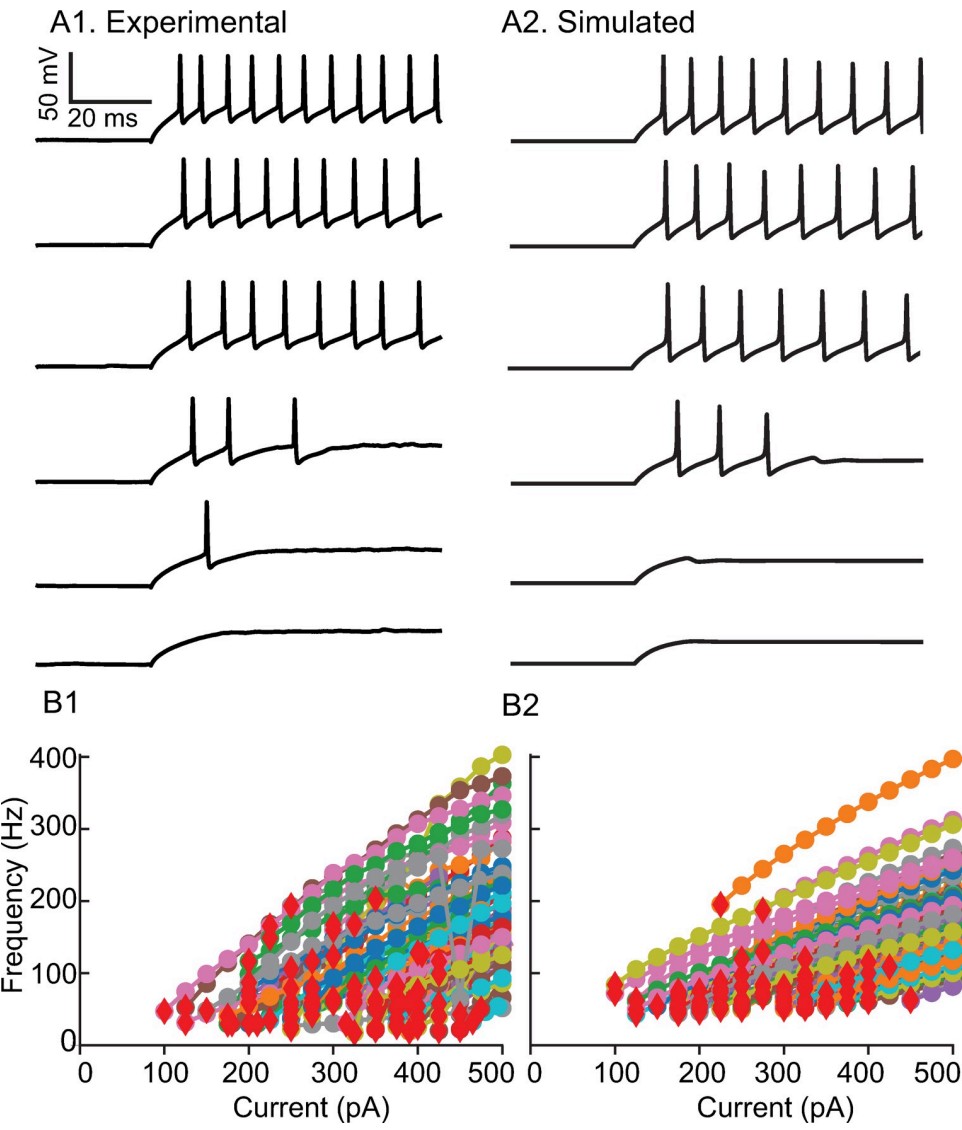

**Fig 1. Calibration of active properties.** A. Responses of representative neurons to depolarizing current steps from 300 pA to 425 pA in 25 pA increments. A1. mEC PV+ interneuron. A2. Model neuron. Parameters as in Table 1 except $E_L$ -80.7 mV, $R_{input}$ 83.55 MΩ ($g_L$ 11.97 nS), $\tau_m$ 5.45 ms ($C_m$ 65.18 pF), $g_{NA}$ 14426 nS, $g_{Kv1}$ 55 nS, $g_{Kv3}$ 709 nS, $\theta_m$ -50.7 mV, $\theta_h$ -53.07 mV, $\theta_n$ 8.79 mV, $\theta_a$ = 49.46 mV. B. Population f-I curves. B1. Experimental. B2. Model.

theoretical insights into the model, we first considered homogeneous networks amenable to analysis via phase response theory under the assumption of pulsatile coupling [27–29]. The homogeneous network consisted of 100 clones of a single model neuron, with identical intrinsic properties, connected through 36 identical presynaptic chemical synapses. The parameters of the single model neuron correspond to one with an f-I curve close to the middle of the heterogeneous range. We incorporated synaptic depression at the chemical synapses, but we did not include gap junctions in order to determine whether the chemical synapses alone could synchronize the network.

Chemical synapses are modeled as GABA_A synapses. Their reversal potential is difficult to measure *in vivo*; it is unclear whether they are hyperpolarizing or shunting (see Discussion). In order to generate testable predictions that differ depending on whether the synapses are

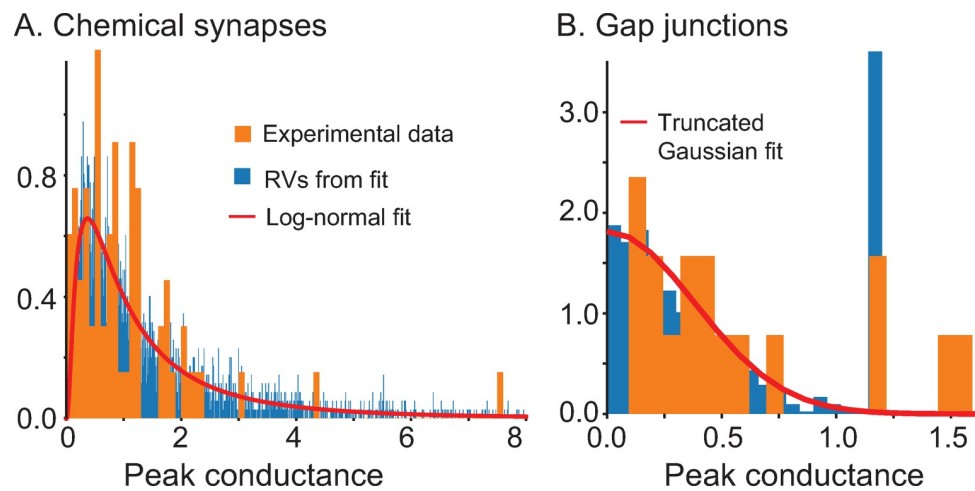

**Fig 2. Calibration of synaptic properties.** Probability densities. A. Chemical Synapses. B. Electrical Synapses.

shunting or hyperpolarizing, we compared model network dynamics using synaptic reversal potentials, $E_{syn}$, values of -75 mV (left column in Fig 3) and -55 mV for hyperpolarizing and shunting synaptic inhibition, respectively (right column in Fig 3). Although shunting networks exhibit faster and more variable frequencies in response to simulated optogenetic sinusoidal drive, both exhibit global synchrony (Fig 3A1 and 3B1). Since the neurons are identical and receive identical input, no connectivity is actually required to synchronize them. Removing synaptic connectivity in a sinusoidally-driven homogeneous network only alters the timing of the population spikes, and not their synchrony (not shown). However, the perfect synchrony present with chemical synapses intact indicates that the synapses themselves are not sufficient to destabilize global synchrony in the presence of a common sinusoidal drive. Adding the full heterogeneity to the intrinsic properties of the neurons in the network as described in Fig 1 completely eliminates the fast oscillations nested in the theta drive (Fig 3A2 and 3B2). In Fig 3A3 and 3B3, we show the response for a network using identical neurons, with heterogeneity only in the synaptic delays, weights, and numbers of presynaptic partners. For both types of inhibition, the height of the first peak is 100, indicating global synchrony for all 100 neurons as in Fig 3A1. However, near perfect global synchrony persists only for the first few population spikes. The mechanisms underlying the decrease in synchrony during the theta cycle will be investigated first with respect to the phase resetting properties of the two types of synapses and subsequently with respect to the effects of synaptic depression.

Next, we used a mean field approach that assumes synchronized oscillations in a homogeneous network in which every neuron is identical and receives identical input (i.e. from exactly

**Table 1. Parameters for gating variables.** The θ parameters are given for the homogeneous network in Fig 4. These parameters were varied across the network in order to reproduce the variability in f/I curves. The other parameters were held constant for all model neurons.

| | *m* | *h* | *n* | *a* |
|---|---|---|---|---|
| Θ (mV) | -53.0 | -55.71 | 5.9 | 51.36 |
| $\sigma_1$ (mV) | 4 | -20 | 12 | 12 |
| $\sigma_2$ (mV) | -13 | 3.5 | -8.5 | -80 |
| $k_1$ (ms$^{-1}$) | 0.25 | 0.012 | 1 | 1 |
| $k_2$ (ms$^{-1}$) | 0.1 | 0.2 | 0.001 | 0.02 |

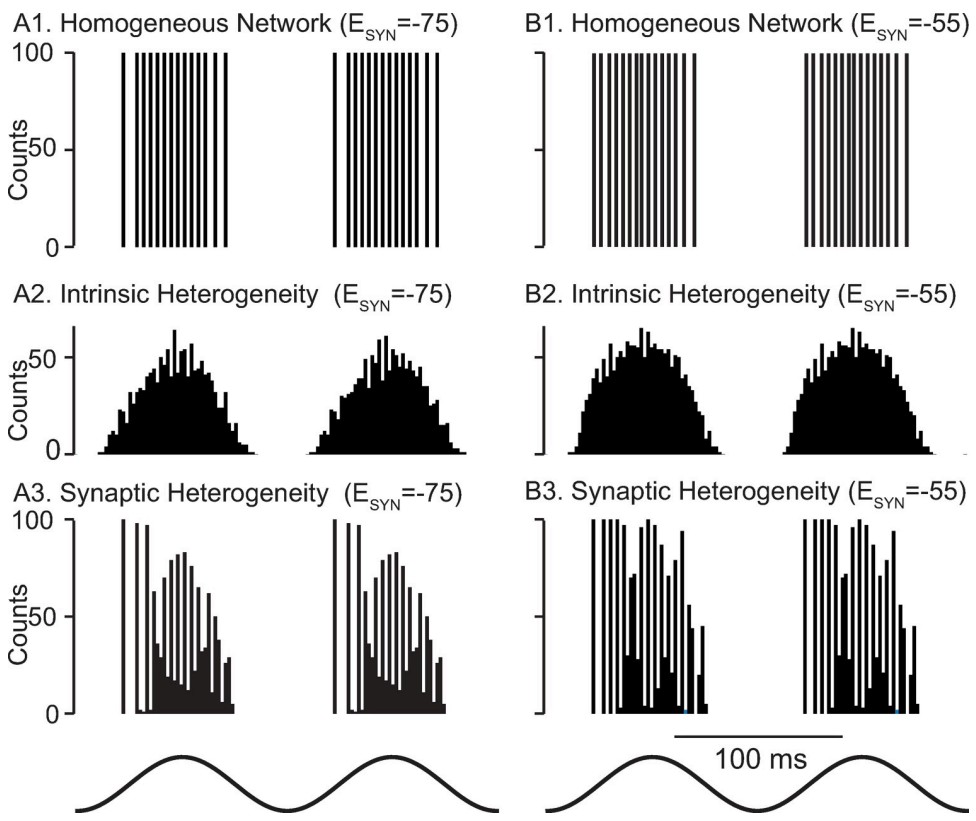

**Fig 3. Biophysically calibrated levels of heterogeneity are desynchronizing.** Representative spike histograms as a function of time. **A1** Hyperpolarizing and **B1** Shunting Homogeneous Networks with these parameters: $g_{NA}$ 16805 nS, $g_{Kv1}$ 59 nS, $g_{Kv3}$ 631.7 nS. $E_L$ -72 mV and $C_M$ 0.0768 nF. $g_L$ was 14.7 nS resulting in an input resistance of 68 MΩ, and $g_{ChR}$ was 7 nS, with others as in Table 1. Each neuron received exactly 36 chemical synapses with a strength of 1.65 nS. The synaptic delay was fixed at 0.8 ms. **A2** Hyperpolarizing and **B2** Shunting Networks with heterogeneity in the active and passive parameters across all 100 neurons. **A3** Hyperpolarizing and **B3** Shunting Networks with homogeneous neurons but randomly instantiated synaptic connectivity, conduction delays and synaptic conductances. The optogenetic theta drive (bottom) varied sinusoidally at 8 Hz from 0 to 14 nS in all panels.

36 other identical neurons). A representative neuron was chosen and cloned 100 times. This allowed us to perform a phase response curve (PRC) analysis to predict the stability and frequency of global synchrony [27–29]. In order to apply phase response theory under the assumption of pulsatile coupling, an autonomous system with constant parameters is required. Therefore, we considered the case of a constant external input at the midpoint of the theta modulation of $g_{ChR}$ (7 nS) in Fig 3. Under these conditions, the intrinsic frequency of the selected model neuron is 168 Hz. We generated a phase response curve (sometimes called a spike response curve [30,31]) by applying an inhibitory postsynaptic conductance on separate trials at each of 100 equally spaced phases within the free-running cycle of the model neuron, with the point at which the neuron reaches threshold (defined as -30 mV) taken as a phase of 0 (and 1). We used a conductance that was 36 times larger than the conductance of a single synapse to simulate the synchronous input received by a single neuron in the network during global synchrony (see inset in Fig 4A). We plotted the normalized increase in the period (the phase delay) as a function of the phase (Fig 4A) for a hyperpolarized synaptic reversal potential (red trace). Hyperpolarizing inhibition (red) consistently induces phase delays that lengthen the period. The two dotted lines indicate the range across which conduction delays (δ) were varied in Fig 3A3. The arrows indicate the phase at which an input would be received in a

## A. Phase Response Curves

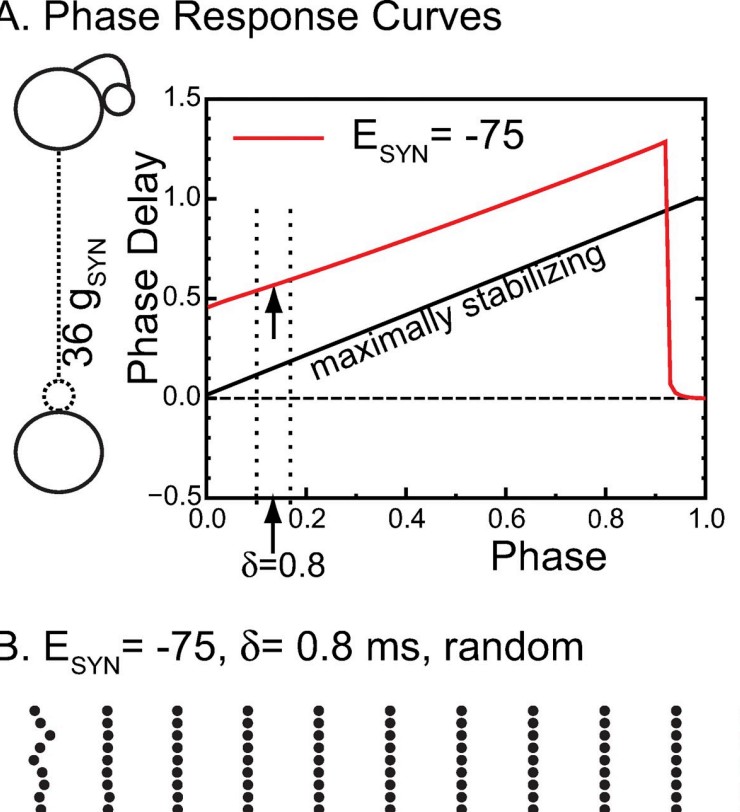

## B. $E_{SYN}$= -75, δ= 0.8 ms, random

**Fig 4. Phase Response Curve Explain Synchronizing Tendencies for Hyperpolarizing inhibition.** A. A biexponential inhibitory postsynaptic conductance was used as the perturbation to a single neuron from Fig 3A1 to generate the PRC for hyperpolarizing (red). The strength of an individual conductance was multiplied by 36 to reflect the 36 simultaneous inputs received by a single neuron (left inset) during perfectly synchronous oscillations. The arrows indicate the phase at which an input delayed by 0.8 ms is received in the network. The dashed lines refer to the range of synaptic delays shown in Fig 3A3 and 3B3. The free running period of this neuron is 5.97 ms at a constant ChR conductance of 7 nS, the midpoint of the excitatory theta drive. B. For hyperpolarizing synapses with conduction delays of 0.8 ms, synchrony is stable and attracts from random initial conditions in a single cycle in this raster plot of 20 representative neurons from the 100 neuron network.

globally synchronous mode at: $\theta = \delta/P_i$, where $P_i$ is the free running period of the neuron. For a conduction delay of 0.8 ms, this phase is 0.168.

In order to apply the phase response curves, an inhibitory input must have the same effect in the network as it did when applied to a free-running neuron at a stabilized frequency in order to generate the PRC in the first place. In practice, the requirement is that the neuron must have returned very close to its unperturbed state (on its limit cycle, speaking mathematically) by the time the next input is received. If the second order phase response [32,33], that is, the change in length in the subsequent cycle, is small, then it is likely that the trajectory has returned close to the limit cycle by the next spike, which in a one-to-one locking precedes the next input. For delays up to 90% of the cycle period, there is less than a 3% change in the length

of the subsequent cycle (not shown). The phase resetting resulting from an input applied at a delay of 0.8 ms predicts a network frequency of 106 Hz for hyperpolarizing synapses. Global synchrony in the homogeneous network is strongly attracting; the network converges to global synchrony in a single cycle after random initialization (Fig 4B). The observed frequency in the homogeneous network is 110 Hz, which is not exact, but is very close to the predicted frequency and illustrates the predictive power of the theory despite some slight deviation from the pulsatile coupling assumption.

Phase response theory can also explain the fast convergence to synchrony. The stability of synchrony is determined by whether a perturbation of even a single neuron from the globally synchronous mode decays or grows on the next cycle. For a synchronous mode with short delays, the perturbation grows or decays according to the scaling factor $1 - f'_{36}(\theta) - f'(\theta)$ [27, 29], where $f'_{36}(\theta)$ is the slope of the phase response curve of the single perturbed neuron at the locking phase (arrow on red curve in Fig 4A) and $f'(\theta)$ is the slope of the phase response curve of the other 99 neurons in response to an input from the perturbed single neuron, also at the locking phase. The other 99 neurons are assumed to receive 35 simultaneous inputs with a delay of 0.8 ms after the population spike, with the perturbed neuron only adding a 36th simultaneous input to the already very strong input, allowing us to neglect $f'(\theta)$ as small compared to $f'_{36}(\theta)$. The expression for the rate at which a perturbation decays then becomes approximately $1 - f'_{36}(\theta)$. The maximally stabilizing value is $f'_{36}(\theta) = 1$ (see illustrative maximally stabilizing diagonal in Fig 4A) since after only a single cycle a perturbation will decay to zero. The slope of the PRC for hyperpolarizing synapses (arrow on red trace) is close to one, which accounts for the rapid convergence in Fig 4B. The sharp decrease in the PRC at late phase occurs because of the finite duration of the waveform of the biexponential synaptic conductance; when it is applied near the end of the cycle, insufficient charge accumulates to delay the action potential substantially; instead, it lengthens the subsequent cycle length. However, if an input is initiated after the action potential has occurred, the cycle containing the start of the perturbation is substantially lengthened. The discontinuities near a phase of 0.9 and between 1 and 0 on the red trace in Fig 4A are highly destabilizing. For global synchrony to be observed in networks with any jitter in the spike times, the conduction delays must be long enough (and short enough) to avoid sampling a discontinuity [34].

The PRC approach is less informative for the network with shunting inhibition because the pulsatile coupling assumption is not well honored. Shunting inhibition decreases the network period, such that the synaptic waveform due to single input persists throughout two or more cycles. Therefore, we plotted the change in cycle length in both the cycle that contains the start of the perturbation (first order PRC, solid green curve in Fig 5A) and in the second cycle (second order PRC, dotted green curve). The slope of the first order PRC is initially negative, which means that very short delays <0.6 ms are destabilized because a negative $f'_{36}(\theta)$ causes the perturbation multiplier $1 - f'_{36}(\theta)$ to have an absolute value greater than one, which leads to growth of the perturbation. Flat PRCs with a slope near zero are only very weakly attracting or repelling. We initialized all neurons but one in the shunting network on their unperturbed, free-running steady limit cycle at the action potential threshold in Fig 5B. A perturbed neuron was initialized at a phase that corresponded to a difference of one tenth of the period relative to the unperturbed neurons. This slight perturbation slowly desynchronized the network, which demonstrates that shunting inhibition at a delay of 0.8 ms (left arrow) is destabilizing for global synchrony. The theta-modulated synchrony in Fig 3B1 for the shunting network likely occurs in spite of, and not because of, the weakly desynchronizing chemical synapses and is therefore driven solely by the common input to identical neurons. Although phase response theory under pulsatile coupling does not strictly apply to the shunting networks, it

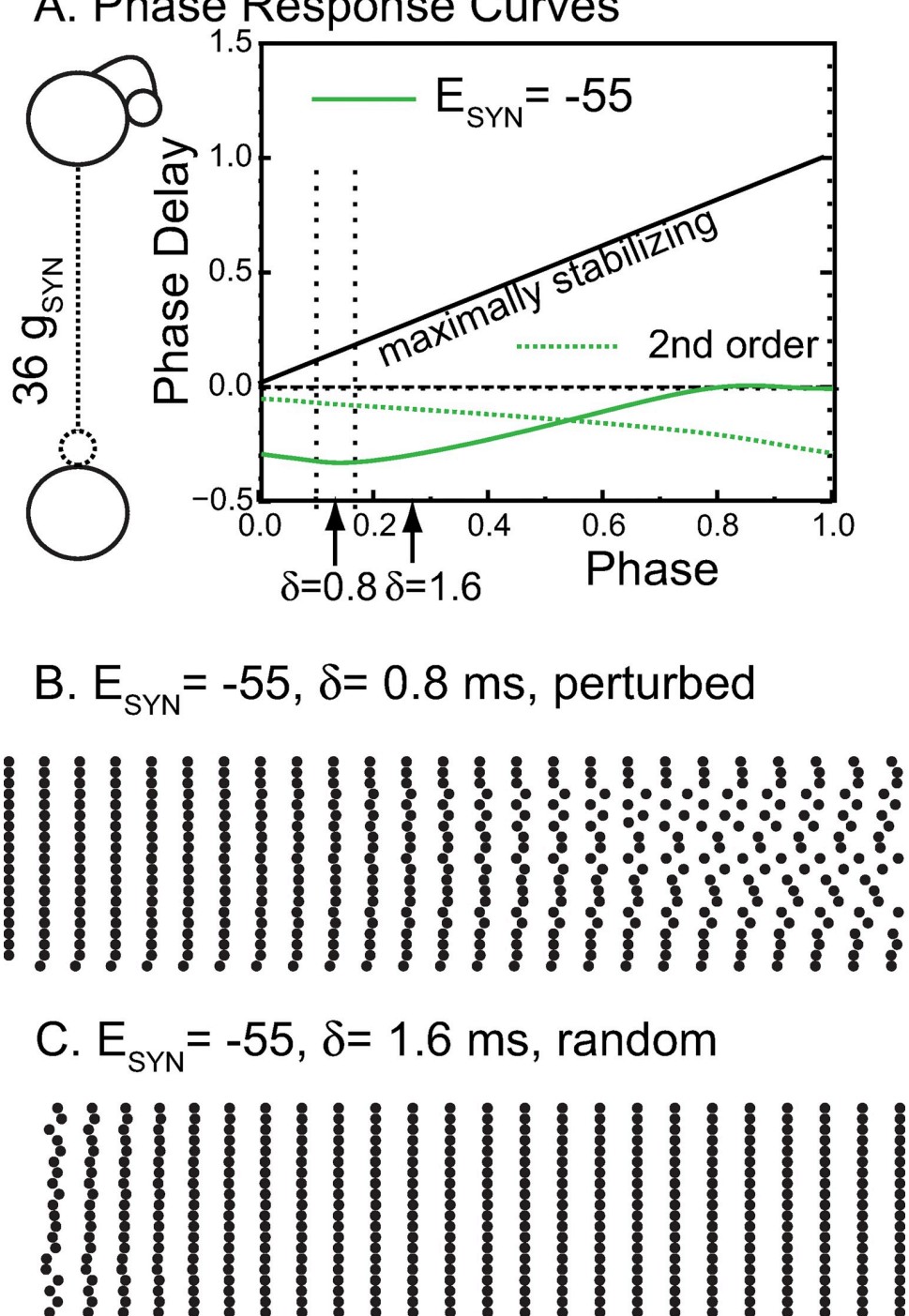

**Fig 5. Phase Response Curve Explain Synchronizing Tendencies for Shunting Inhibition.** A. A biexponential inhibitory postsynaptic conductance as the perturbation to a single neuron from Fig 3B1 to generate the PRC for shunting inhibition (green). The dashed green curve shows the normalized change in the cycle after the cycle that contains the perturbation (second order). The strength of an individual conductance was multiplied by 36 to reflect the 36 simultaneous inputs received by a single neuron (left inset) during perfectly synchronous oscillations. The leftmost arrows indicate the phase at which an input delayed by 0.8 ms is received in the network. The dashed lines refer to the range of synaptic delays shown in Fig 3A3 and 3B3. The free running period of this neuron is 5.97 ms at a constant

ChR conductance of 7 nS, the midpoint of the excitatory theta drive. B. For shunting synapses, starting from exact synchrony, perturbing even a single neuron (bottom trace) eventually desynchronizes the network. C. If the conduction delay is increased to 1.6 ms in the network with shunting inhibition, synchrony is stabilized and attracts quickly from random initial conditions. B-C are raster plots of 20 representative neurons from the 100 neuron network. Parameters are as in Fig 3 except for $E_{SYN}$.

can still provide some insights. For example, it suggests that the network will synchronize if the conduction delay is increased from 0.8 ms to 1.6 ms (rightmost arrow on solid green curve), where the slope is closer to one and more strongly synchronizing. Indeed, synchrony arises from random initial conditions (Fig 5C) when the conduction delay is set to 1.6 ms. The convergence, however, is not as fast as in Fig 4B, which is likely due to the slope being farther from one. The observed frequency of 241.5 Hz is again similar to the predicted frequency of 247 Hz. The first order PRC for shunting, but not hyperpolarizing, inhibition reverses sign at a phase of 0.14, very near the action potential trough at a phase of 0.175. This suggests that action potential width, along with conduction delay and synaptic rise time, may be a determinant of synchronization tendencies for shunting, but not hyperpolarizing inhibition in the mEC.

Recent work in neocortex found that a significant fraction of the inhibition received by PV + interneurons in that area was mediated by autapses [35]. The synchronization properties of both hyperpolarizing and shunting networks were unchanged by shifting a third of the inhibition from synapses from other interneurons to autapses (not shown).

## Response of heterogeneous networks with hyperpolarizing versus shunting synapses to theta modulation: effects of gap junctions and synaptic depression

We showed in Fig 3A2 and 3B2 that the full complement of observed heterogeneity in the intrinsic properties of the model by itself suppressed theta-nested fast oscillations. Therefore, it is unsurprising that in the presence of full heterogeneity (both intrinsic and synaptic), theta-nested fast oscillations are also suppressed in networks with either hyperpolarizing or shunting inhibition (Fig 6A1 and 6B1). However, the heterogeneous networks in Fig 3A2, 3A3, 3B2 and 3B3 neglected gap junctions in order to assess the effects of chemical synapses in isolation. In fact, gap junctions between PV+ interneurons in the mEC are highly prevalent [15], therefore we tested their impact in the model. As described in the Methods, the experimentally recorded connection probability and distribution of gap junction peak conductances suggest that they make a substantial contribution to the measured input resistance. Our initial calibration of f-I curves in Fig 1A2 and 1B2 ignored gap junctions; therefore, incorporating gap junctions required recalibration of the passive properties of the model neurons. As described in the Methods, we reduced the leakage conductance and adjusted the reversal potential each time a gap junctional conductance was added to a model neuron in order to preserve the original range of values for the input resistances and resting membrane potentials. A minimum value for the leak conductance, $g_L^{min}$, was set at 1.5 nS to honor the constraint that the interneurons must have at least some intrinsic leak conductance. Although imposing this constraint decreased the total number of electrical synapses, the effect was mitigated because the exact number of synaptic contacts of either kind is not known in the mEC, as noted in the Methods. The parameters for voltage-gated ion currents were not changed, and f/I curves were minimally affected by adding the gap junctions (S2 Fig).

For networks with hyperpolarizing chemical synapses, when physiologically constrained gap junctions were added as described above, network synchrony at fast frequencies increased during the theta cycle (Fig 6A2), then decreased. Fast oscillations were not visible for

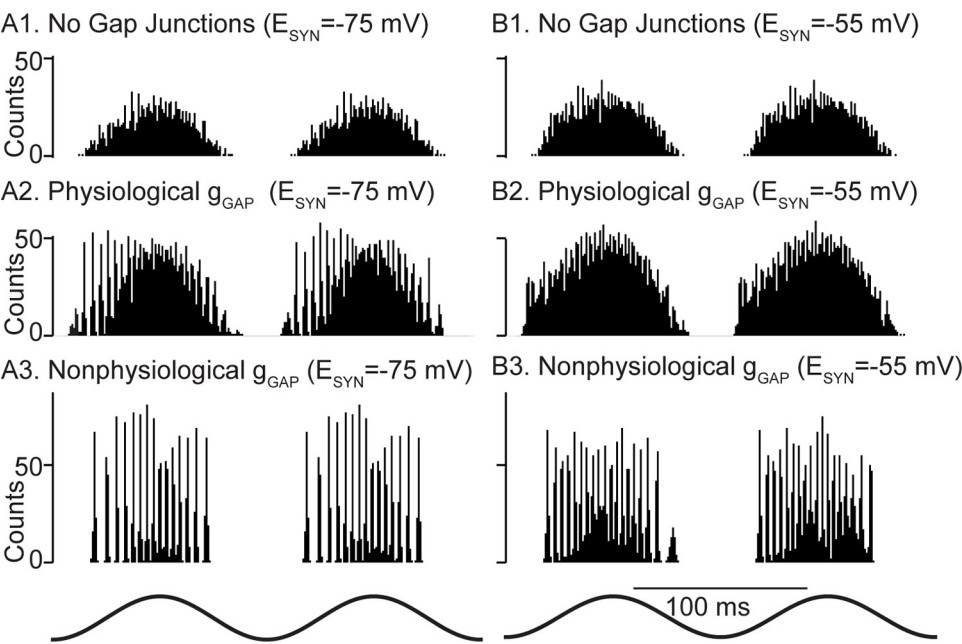

**Fig 6. Gap junction connectivity is required for theta nested fast oscillations in heterogeneous networks.**
Representative spike histograms as a function of time. **A.** Networks with hyperpolarizing inhibition. **B.** Networks with
shunting inhibition. **A1.** and **B1.** Fully heterogeneous networks (both intrinsic and synaptic heterogeneity) with no gap
junctions. **A2** and **B2**. Heterogeneous networks with gap junctions calibrated according to Fig 2B and the text
accompanying this figure. **A3** and **B3**. Heterogeneous networks with 2 nS gap junctions with no compensatory
reduction in leakage conductance.

physiological gap junction connectivity with shunting chemical synapses (Fig 6B2). Strong
homogeneous gap junction strengths (2 nS) resulted in tight synchronization for both hyper-
polarizing and shunting inhibition (Fig 6A3 and 6B3). Under these conditions, however, it was
not possible to compensate for such strong gap junctional conductances by decreasing the leak
conductance while still maintaining input resistance values within the experimentally con-
strained ranges. Also, this strength is beyond the physiologically observed range in Fig 2B. We
deemed this gap junctional connectivity non-physiological for those two reasons. The greater
robustness of global synchrony of theta-nested fast oscillations was preserved at the larger
value (1.6 ms) of conduction delay (S3 Fig), despite the synchronizing tendency of networks
with shunting synapses already detected at that value for homogeneous networks in Fig 5C.

In order to demonstrate that the results in Fig 6 were not specific to one random connectiv-
ity pattern, we constructed 30 networks that differed in their connectivity pattern in both
chemical and electrical synapses. In addition to distinct connectivity graphs, the peak conduc-
tances for the two types of synapse and the delays for the chemical ones were obtained from a
different sampling of their respective distributions. In Fig 7A, each filled dot represents a dif-
ferent network, and the physiological level of gap junctional connectivity was implemented as
in Fig 6A2. Physiological levels of gap junctions were effective in stabilizing networks using
hyperpolarizing synapses (Fig 7A) but not shunting synapses, regardless of the connectivity
pattern. No evidence for nested fast oscillations was found for shunting networks; hence, there
is no corresponding plot for that case. The frequency of the nested oscillations is relatively sta-
ble across network instantiations for hyperpolarizing synapses (Fig 7A). The different degrees
of synchrony for different connectivity patterns, as measured by the maximum power, suggest
that higher order statistics in the connectivity graph, like the presence of hubs or loops, could

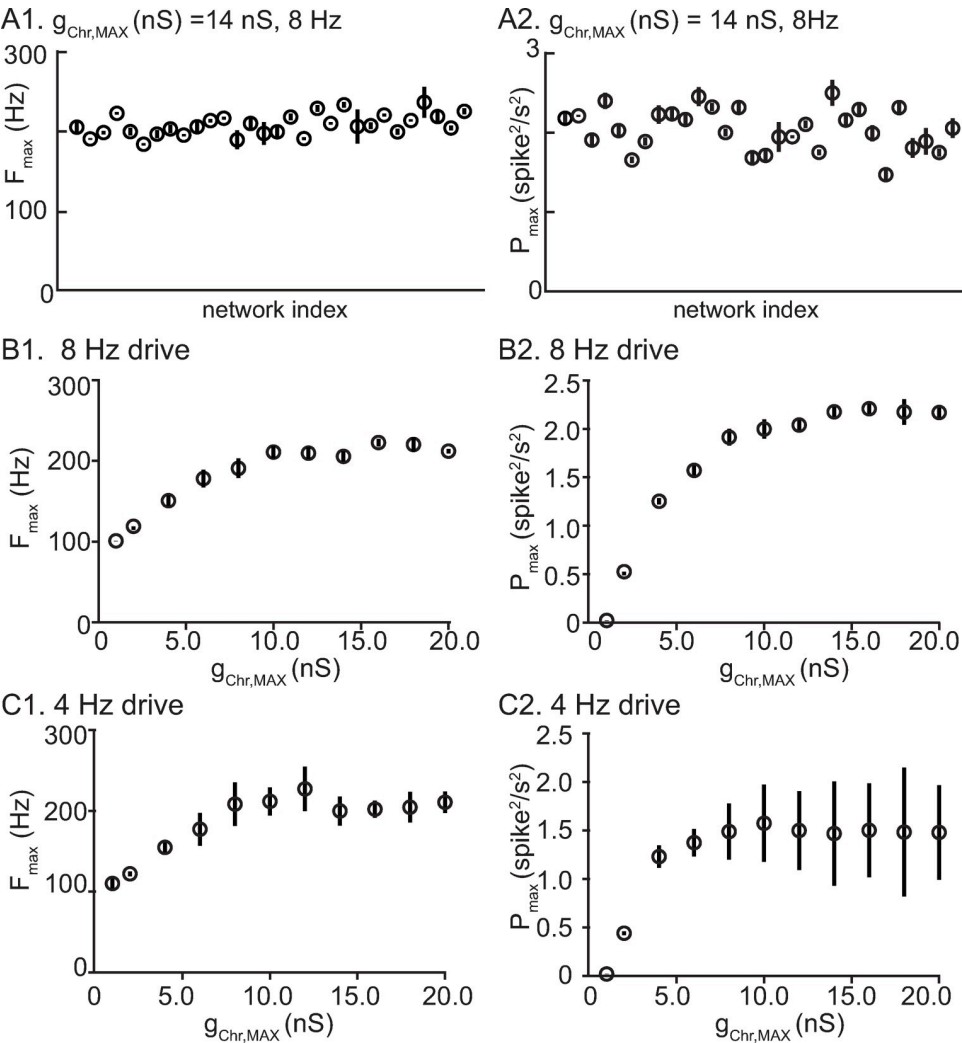

**Fig 7. Performance of network with hyperpolarizing synapses is robust to different instantiations of network connectivity. A**. Distinct random network instantiations with constant drive amplitude and frequency. Vertical lines give the standard deviation within a network across theta cycles. **A1**. Fast frequency with the most wavelet power for each network. **A2**. Maximum power. **B**. Same network, variable amplitude 8 Hz drive. **B1**. Frequency with max power. **B2**. Maximum power. **C**. Same network, variable amplitude 4 Hz drive. **C1**. Frequency with max power. **C2**. Maximum power. For B and C, vertical lines give the standard deviation for 15 network instantiations across all theta cycles.

also enhance or hinder synchrony, especially for the slower drive. Alternatively, networks in which similar neurons are more strongly connected might be more predisposed to synchrony.

Next we examined the effect of varying the amplitude of the theta drive at the same frequency as in the previous panel (Fig 7B) and at a slower frequency, 4 Hz (Fig 7C). The frequency at which the maximum power was observed saturated at about 200 Hz for both driving frequencies, and the maximum power saturated as well. During the rising phase of the theta drive, if the amplitude of drive reaches a level that allows a network frequency of about 200 Hz to be attained, the network desynchronizes as described in the next paragraph. The frequency with maximum wavelet power occurs just prior to desynchronization. Many manipulations may decrease (increase) the frequencies relative to a fixed theta drive, such as increasing (decreasing) the synaptic decay time, increasing (decreasing) the time constant of individual neurons, or slowing (speeding) the deactivation of $K_V 3$. However, the maximum network

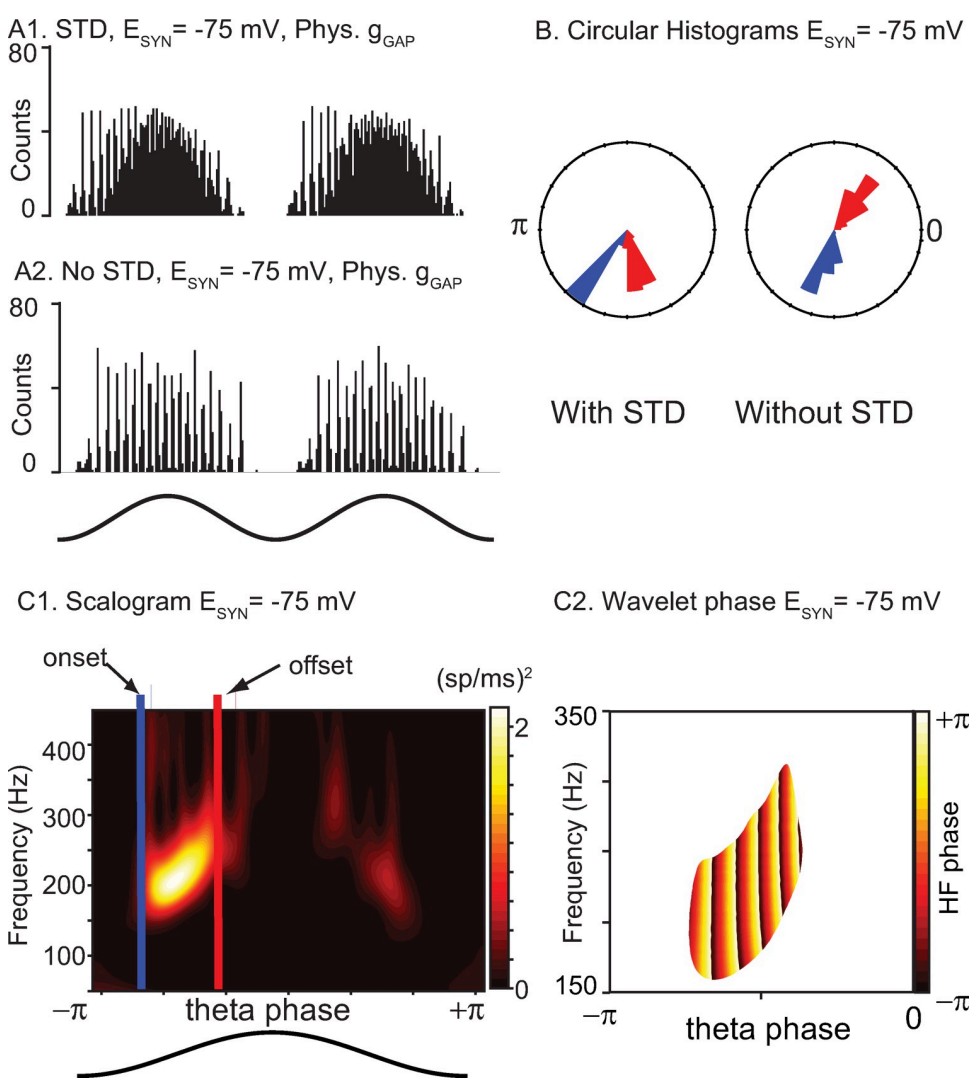

**Fig 8. STD induces a preference for theta phases before the peak in networks with hyperpolarizing synapses.** A1. Repeated from Fig 6A2, theta-nested fast oscillations increase on the rising phase of theta stimulation but decrease after the peak due to synaptic depression. A2. Removing short-term depression from the network restores symmetry about the peak for hyperpolarizing networks. B. Circular histogram of onset (blue) and offset (red) phases with and without STD. C. Wavelet analysis. C1. Scalogram of power at each frequency showing how onset and offset phases were determined. C2. Wavelet phase between onset and offset for the bright region of high power bracketed between the blue and red bars in C1. The x and y axis were rescaled to emphasize the region containing theta-nested high frequency oscillations.

frequency, followed by desynchronization, is simply reached earlier or later within the theta cycle. In the absence of STD, slightly higher frequencies can be achieved (250 Hz) before the network desynchronizes; the inhibition from successive population spikes summates at high frequencies and becomes more tonic and less phasic [17], likely favoring desynchronization.

We next examined the effect of short-term depression (STD) on the model networks by removing STD from the model. For networks with hyperpolarizing synapses, we previously showed (Fig 6A2) that they preferentially exhibited fast oscillations on the intracellular rising (extracellular falling) phase of theta stimulation compared to the falling phase. Removing STD clearly negated this phase preference (compare Fig 8A2-A1) and rendered the fast oscillation amplitude symmetric about the peak (extracellular trough). Since STD decreases the overall

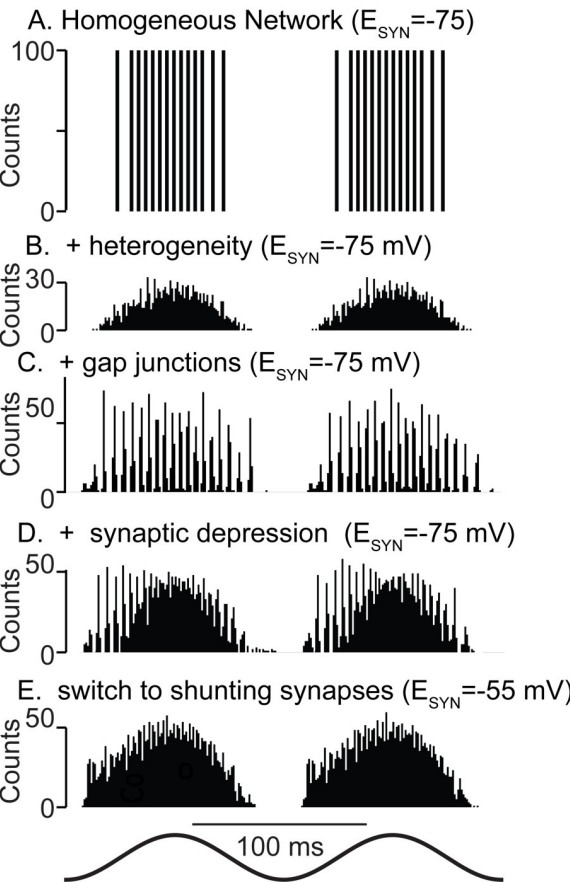

**Fig 9. Summary Figure. A.** Theta drive (bottom trace) synchronizes homogeneous networks provided there is a minimum conduction delay. **B**. Full heterogeneity disrupts synchrony. **C**. Adding physiological levels of gap junctions restores synchrony. **D**. As hyperpolarizing synapses depress during the theta cycle, synchrony is lost. **E**. Shunting synapses do not synchronize the fully heterogeneous system even at early phases in the presence of synaptic depression and gap junctions.

contribution of the chemical synapses, we conclude that in our networks, hyperpolarizing inhibition increases the robustness of synchrony at fast frequencies. In contrast, shunting inhibition weakly opposes synchrony, and its effect is not shown here because its impact is minimal when chemical synapses are calibrated as described in the text accompanying Fig 2. In the presence of very strong gap junctions, the tendency to oppose synchrony is only revealed by increasing the strength of the shunting conductance fivefold (S4 Fig), which greatly decreases the power of the nested fast oscillations. Fig 8B1 shows a scalogram for a representative network with STD showing that the power is concentrated in the 150–200 Hz range. The onset and offset of fast oscillations were computed from the phases at which the power crossed from below and from above, respectively, a threshold of 0.3 times the maximum power across the 30 cycles of the simulation. The nesting of fast oscillations within theta is evident in the plot of the wavelet phase (Fig 8C2) only in the region of high power shown in Fig 8C1. The circular histograms in Fig 8B show these onset (blue) and offset (red) phases pooled across 30 simulated theta cycles of all 30 simulated networks with different connectivity patterns. The case with STD is shown on the left, and without STD on the right. The offset phase (in radians) with and without STD was -1.33±0.13 and -0.90±0.18, respectively, with zero being the peak of the theta

stimulation. Further, the onset phase with and without STD was -2.21±0.07 and -1.81±0.17, respectively. The theta phase offset was substantially and significantly different (p<0.001) using Watson's U2 test in the circular statistics package in R (CRAN, RRID:SCR_003005) [36]. The theta phase onset differed only slightly between the two conditions; however, this difference was also significant (p<0.001). The smaller range of theta phases that support nested fast oscillations with STD compared to without them supports the premise that hyperpolarizing inhibition helps to synchronize inhibitory networks in the presence of biological levels of heterogeneity, provided there is also a biological level of gap junctional connectivity.

## Discussion

### Summary

We used an experimentally calibrated computational model of a network of fast-spiking parvalbumin-positive inhibitory basket cells (PVBCs) to study their synchronizing properties, as well as the properties of the emerging oscillations and the underlying mechanisms. The model was calibrated using electrophysiological recordings from mouse mEC slices and reproduced the full range of heterogeneity in the experimental f/I curves, including their high cutoff frequencies, which indicate type II excitability. We calibrated both the neural intrinsic passive and active properties. The former includes the leakage reversal potential, leakage conductance and membrane time constant. The latter comprise the parameters that determine the dynamics of voltage-gated ion currents. We also used recordings to calibrate the properties of both chemical and electrical synapses between the neurons. Network behavior was studied using a theta-modulated input current simulating a channelrhodopsin-driven optogenetic input similar to previous studies [12–14].

The main results are summarized in Fig 9, namely that heterogeneity can destroy synchrony (compare Fig 9A and 9B), that gap junctions can rescue synchrony (Fig 9C), and that a preference (Fig 9D) for fast oscillations in the rising phase of excitatory theta drive (descending phase of extracellular theta) is a hallmark of hyperpolarizing inhibition when combined with the short-term synaptic depression commonly observed in synapses made by PV+ neurons [37,38], with the caveat that a minimum amplitude for the theta drive is required to manifest this phase preference. This phase preference is consistent with that observed in PV+ basket cells in CA1 during theta-nested ripples [39]. Moreover, we find that shunting inhibition desynchronizes networks with the synaptic and intrinsic properties characteristic of the mEC (Fig 9E). Our results strongly suggest that any ING in mEC is mediated by hyperpolarizing rather than shunting inhibition, and that artificially manipulating the synaptic reversal potential to make it more depolarizing should decrease optogenetically-evoked ING that persists after blocking excitatory synapses. As in a previous study [40], we observed synergy between gap junctions and hyperpolarizing inhibition. Gap junctions mitigate the effect of heterogeneity by forcing the activity of different neurons to be more similar than if gap junctions are absent. Further predictions are that gap junctional connectivity is required for the expression of fast oscillations mediated by inhibitory interneurons in the mEC, and that blocking gap junctions would not only disrupt synchrony, but also change the measurable input resistance and time constants of isolated PV+ neurons. This is consistent with previous estimates [41] in which gap junctions account for one third to one half of the input conductance of fast spiking interneurons, and in some cases more than half [42]. Unfortunately, blocking gap junctions selectively is challenging experimentally as the gap junction blockers are non-specific and can block voltage-gated $K^+$ currents, which limits the ability to discern the specific impact of gap junctions on input resistance [43].

## Shunting versus hyperpolarizing synapses

Here, we modeled chemical synapses between PV+ cells as ionotropic GABA$_A$ receptors. Whether inhibition is shunting or hyperpolarizing depends upon the chloride reversal potential, as well as on the reversal potential for bicarbonate ions flowing in the opposite direction [44,45]. Larger contributions of bicarbonate lead to more depolarized synaptic reversal potentials, and these contributions may vary between brain regions. Early studies in CA1 and CA3 found hyperpolarizing inhibition between basket cells [17,18]; in contrast, the inhibition between fast spiking basket cells in the dentate gyrus *in vitro* is shunting, with a reversal potential of about -52 mV [16]. Moreover, the intracellular Cl- concentration is not static and can be modulated; for example, activation of kainate-type glutamate receptors potentiates the activity of the potassium-chloride co-transporter 2 (KCC2) via interactions with the GluK2 subunit, reducing the intracellular chloride concentration and rendering the reversal potential more hyperpolarized [46]. There is also evidence that steady-state intracellular Cl$^-$gradients within neurons may be set by cell-type-specific, subcellular expression patterns of functional cation chloride cotransporters [45]. As a result of all these factors, there is sufficient uncertainty regarding the precise reversal potential of GABA$_A$ synapses between interneurons to warrant our systematic study of both types of inhibition.

## Fast oscillations in the entorhinal cortex

At least two types of fast oscillations have been observed in the entorhinal cortex: fast gamma oscillations in the 65–140 Hz range [7] and ripples 140–200 Hz [47]; they can coexist *in vitro* when induced by kainate application [48]. The frequency border between these oscillations is ambiguous in the literature; some quote 100 Hz as the border [49], whereas others label 90–150 Hz as an epsilon band or alternatively refer to the 65–90 Hz range as medium gamma and 90–140 Hz as fast gamma [50]. However, gamma oscillations are frequently nested in theta oscillations [51], whereas ripples are often nested in sharp waves [52]. The frequencies observed in our carefully calibrated model of layer 2/3 mEC PV+ interneuronal networks consistently fall in the upper end of that range (~150–200 Hz). This value is consistent with the frequency of fast oscillations evoked in a study that selectively activated PV+ neurons at theta frequencies [14]. It is possible that the high frequency oscillations observed in that study and in our model are more analogous to the ripples in superficial mEC that contribute to ripple bursts and extended replays in area CA1 in quiet awake rodents [47]. Our study focused on ING and did not include excitatory neurons; perhaps *in vivo* neural populations such as the stellate, pyramidal or other inhibitory neurons contribute to slowing the oscillations into a fast gamma range via synaptic or modulatory mechanisms. It has been previously suggested [53] that fast gamma oscillations are more similar to ripples than to slow gamma, and result from interneurons escaping control of phasic excitation and entering a regime of tonic excitation. In the former, the interneurons do not fire unless prompted by the phasic excitation, hence their intrinsic dynamics only contribute to setting the frequency in the tonic excitation regime. This interpretation is not universally accepted, however, as an alternative hypothesis posits that the ripples are merely transients resulting from a strongly synchronizing input [54].

## Relationship to previous models of fast oscillations in heterogeneous inhibitory interneuronal networks of coupled oscillators

Synchrony mediated by inhibition was originally thought to require a synaptic rise time longer than the duration of an action potential [55]. A pioneering study on the effects of heterogeneity in networks of interneurons generating fast oscillations implemented heterogeneity by simply

changing the bias current of model neurons that were otherwise identical [11]. In the absence of conduction delays, synchrony was only observed for hyperpolarizing (but not shunting) synapses, and even then only for modest levels of heterogeneity in the bias current. A subsequent computational study [16] also implemented heterogeneity through the bias current; this study included conduction delays, and used stronger and faster synapses compared to the earlier study. Under these conditions, they found that shunting inhibition conferred greater robustness of synchronization at gamma frequency to heterogeneity in the excitatory drive than hyperpolarizing inhibition. Neurons with different levels of bias current traversed a different range of membrane potentials during the interspike interval (the more depolarized the bias current, the more depolarized the envelope). The effect of synaptic input with a shunting reversal potential was therefore different for different single interneurons. Specifically, it caused a phase advance in the slower, more hyperpolarized neurons that increases the spike frequency, while leading to a phase delay when applied to the faster, more depolarized neurons that lowers the frequency. Therefore, shunting inhibition homogenized the rates, pulling them toward the center of the range. Although this study focused on interneuronal models with type 1 excitability, they reached similar conclusions using type 2 model neurons [56]. Our previous work on theta-nested gamma oscillations in inhibitory networks [57] also implemented heterogeneity using different levels of bias current; we found tighter phase locking of gamma oscillations to the theta modulation in type 2 models when using hyperpolarizing inhibition, and in type 1 models with shunting inhibition. Whether hyperpolarizing or shunting inhibition is more synchronizing and potentially more robust to heterogeneity will likely to depend on the exact model and synaptic parameters.

## Phase response curves

Phase response curve (PRC) analysis provides a potential mechanism by which conduction delays (or equivalently, slow synaptic rise times) can stabilize synchrony by specifically avoiding discontinuities in the PRC. Phase responses to strong inhibition often contain destabilizing discontinuities near a phase of 0 (or 1) [21,34,58,59], see Fig 5A (solid red and green traces). A conduction delay of sufficient duration can prevent noise from causing neurons to receive inputs on opposite sides of the discontinuity and stabilizes synchrony [27,34]. In addition, in some cases, there may be an initial region of negative, destabilizing slope (Fig 5A, solid green trace at early phases). The region is destabilizing because if noise speeds up a neuron, accelerating its trajectory such that it receives an input from the population at a phase later than the 1:1 locking phase, the input further speeds the trajectory by advancing the time of the next spike. A slope of one is maximally stabilizing in this scenario because the phase response exactly compensates for the input arriving later (earlier) by delaying (advancing) the spike by exactly the difference between when the input actually arrived, and when it would have arrived in a synchronous mode [28,29,60]. The slope of the phase resetting curve at the locking point can also provide insight into the speed at which the network will synchronize; if it is flat, synchrony can only be weakly attracting and easily disrupted by noise. Another insight is that a steeper PRC has a larger range of advances and delays available so that it is more likely to be able to adjust its own frequency to match that of the population.

## Neural mass models and the mean field approach

Recently, elegant, low dimensional neural mass models have been described that accurately capture the fluctuation in rate during nested gamma of networks of quadratic-integrate-and-fire models driven at theta [61], and which also capture some aspects of optogenetically-driven theta nested gamma in hippocampal area CA1 [62]. One very interesting finding from [61] is

that a theta phase preference, opposite to the one created by synaptic depression in our study, was observed as the theta forcing drives the neural mass model through a subcritical Hopf bifurcation, resulting in a bias for nested gamma towards later, rather than early, theta phases. These models are based on some simplifying assumptions. First, the dynamics of the component neurons can be captured by assuming they are one-dimensional integrators of current. Second, the inputs presented to them are current inputs that summate linearly. Third, the heterogeneity in the network is captured by giving neurons different bias currents. Fourth, that the connectivity is all to all. Fifth, synaptic plasticity is ignored. The last two constraints have been respectively relaxed in recent neural mass models [63,64], but the first three remain intractable to mass models. In our model, the component neurons are resonators [65] with Hodgkin's type 2 excitability [18] rather than integrators with type 1 excitability [19]. The chemical and electric synaptic inputs and the external theta drive are not currents, but rather are conductance based. Conductance based inputs do not sum linearly and will saturate. The inclusion of conductance based synapses in neural mass models [66] on the premise that the neuron is generally at or near its rest potential, thus making the current proportional to the synaptic reversal potential, still assumes that conductances add linearly. In the neural mass models, the different bias currents ensure that the same external input will cause different neurons to be biased at different points along their identical f/I curves. However, the heterogeneity we described here is far more complex and desynchronizing than simple differences in bias current in identical neurons. Our networks contain heterogeneity in the intrinsic passive, intrinsic active and synaptic conductances and connectivity, as well as in the kinetics of the active currents.

Another approach to neural mass models is to use a pulse-coupled phase oscillator model [67] in which each neuron is represented by a phase model. The phase advances at a constant rate determined by the intrinsic frequency when the neuron is unperturbed, but inputs from other neurons advance or delay the phase. A neural mass model based on this strategy [68–70] cannot be applied to our networks because they assume an infinitesimal PRC of constant sinusoidal shape. This is not consistent with conductance-based synapses in which the shape of the PRC depends on the conductance strength [28]. Moreover, for non-instantaneous synapses, the PRC is undefined in the case of an input received before the effect of the previous input has dissipated because the phase is only defined on the limit cycle.

Our mean field approach in Figs 4 and 5 is intended to provide insight into the types of manipulations that affect neural systems whose dynamics cannot be known in sufficient detail to model precisely. The only case for a large network that we know how to treat exactly is a homogeneous system in which all neurons can be assumed to fire simultaneously. The pulse-coupled PRC analysis is a vast simplification of the model network, which in turn is a simplification of the experimental preparation. However, it does explain why conduction delays are required for synchrony for strong inhibition [34]. It also suggests that the steepness of the PRC contributes to the more highly synchronizing effects of hyperpolarizing versus shunting inhibition.

## Caveats on generality

Our study is specific to layer 2/3 medial entorhinal cortex because both intrinsic and synaptic parameters of the network were constrained by data from this region [15]. The only other study that has previously attempted to capture the true extent of heterogeneity in the intrinsic properties of the PV+ interneuronal network, carried out in the external segment of the globus pallidus [71], did not address network synchronization. A recent model of the PV+ interneuronal network in the dentate gyrus considered both heterogeneity in the synaptic connectivity, as

well as the Poisson excitatory synaptic drive input, and used combinations of five different reconstructed interneurons in networks with 200 interneurons [72]; they found that the dendritic location of heterogeneous excitatory input greatly mitigated its desynchronizing effects, regardless of whether the inhibition was shunting or hyperpolarizing. Although we have strong evidence for type 2 excitability of PV+ neurons in the mEC [21,24], previous work supports type 1 excitability in the globus pallidus [73], substantia nigra pars reticulata [74], dentate gyrus [75] and hippocampal area CA1 [11,76,77], which may partially account for disparate results. Moreover, the PV+ neurons in both the globus pallidus and the substantia nigra pars reticulata are spontaneous pacemakers in a slice preparation [78], whereas PV+ neurons in the other regions are quiescent. The synchronization properties of different brain circuits containing inhibitory PV+ networks are not generic and likely differ greatly between brain regions.

Another difference between brain regions is the short synaptic delays (between 0.6–1.0 ms) observed in the mEC, which correspond well to an axonal arbor with 250 μm radius in L2 of the mEC [24], assuming a floor of about 0.4 ms for the synaptic delay and a conduction velocity on the order (0.5 m/s). The short range of connectivity of basket cells is a feature of the mEC which contrasts with the wide axonal arbors of basket cells (500–1740 μm) in the hippocampus [79] and dentate gyrus [80]. Perhaps this feature contributes to highly localized synchrony involved in mEC grid cell assemblies [12].

## Coupled oscillators versus stochastic population oscillators

Two modes of neural synchrony have been proposed [49,81,82]: a strong synchrony in which coupled oscillators fire on every cycle, and a weak synchrony that arises from the population dynamics in which the firing of individual neurons is sparse and appears stochastic. For strong synchrony, most if not all of the recruited neurons fire on almost every cycle of the network oscillation, and the interspike interval histogram has a sharp peak at the network frequency, possibly with subharmonic peaks indicating skipped cycles. For weak synchrony, neurons fire sparsely and irregularly with only a few neurons participating in any given cycle of the network oscillation. Despite clear peaks in the spike density at the population period, the sparseness can obscure any peaks in the ISI histogram of individual neurons such that it resembles a left-truncated exponential distribution characteristic of a Poisson process with a refractory period.

We are not aware of single unit recordings in the mEC during ripples, but in area CA1, PV + neurons fire at 122±32 Hz [83] during ripples *in vivo*, with PV+ basket cells discharging on virtually every ripple event [39,84,85]. Such high firing rates are clearly not consistent [8] with a stochastic population oscillator [82], and our model in this study of theta-nested fast oscillations is clearly a coupled oscillator model. A recent computational study [86] on ripple generation in area CA1 found that, in some cases, an inhibitory interneuronal population can exhibit strong synchrony, while the excitatory neuron population simultaneously exhibits weak stochastic synchrony. That study modeled single neurons as conductance-based leaky integrate-and-fire neurons and assumed that fast oscillations are a network dynamical pattern that does not crucially depend on the details of subthreshold dynamics and spike generation. In contrast, we hypothesize that the details of subthreshold dynamics and spike generation crucially affect synchronization via their phase response tendencies, which can differ greatly from those of leaky integrate and fire neurons [49].

## Methods

### Ethics statement

All experimental protocols were approved by the Boston University Institutional Animal Care and Use Committee.

## Transgenic mice and slice preparation

C57BL/6J background, PV-Cre mice [87] (Jackson Labs, stock # 017320) were crossed with the lox-stop-lox tdTomato reporter mice [88] (Jackson Labs, stock # 007914) to visualize PV + interneurons. Horizontal slices of entorhinal cortex and hippocampus were prepared from 2–8 month-old mice of either sex. After anesthetization with isoflurane and decapitation, brains were removed and immersed in 0˚C sucrose-substituted artificial cerebrospinal fluid (in mM): sucrose (185), KCl (2.5), $NaH_2PO_4$ (1.25), $MgCl_2$ (10), $NaHCO_3$ (25), Glucose (12.5), $CaCl_2$ (0.5). Recordings were taken from slices between 3.2 mm and 4.3 mm from the dorsal surface (bregma) of the brain. Slices were cut to a thickness of 400 μm (Leica VT 1200, Leica Microsystems). Slices were then incubated at 35˚C for 20 minutes in artificial cerebrospinal fluid (ACSF) consisting of the following (in mM): NaCl (125), $NaHCO_3$ (25), D-glucose (25), KCl (2), $CaCl_2$ (2), $NaH_2PO_4$ (1.25) and $MgCl_2$ (1). Afterwards, slices were cooled to room temperature (20˚C). After the incubation period, slices were moved to the stage of a two-photon imaging system (Thorlabs) with a mode-locked Ti:Sapphire laser (Chameleon Ultra II; Coherent) set to wavelengths between 915 nm and 950 nm, which was used to excite both the Alexa Fluor 488 and tdTomato. The stage of the microscope contained recirculating ASCF, with all recordings conducted between 34˚C and 36˚C.

## Electrophysiology

Electrodes were pulled using a horizontal puller (Sutter Instruments) using filamented, thin-wall glass (Sutter Instruments). Intracellular pipette solution consisted of the following (in mM): K-gluconate (120), KCl (20), HEPES (10), diTrisPhCr (7), $Na_2ATP$ (4), $MgCl_2$ (2), Tris-GTP (0.3), EGTA (0.2) and buffered to pH 7.3 with KOH. To visualize electrodes, the cyan-green fluorescent dye Alexa Fluor 488 hydrazide (Thermo Fisher Scientific) was added to the intracellular electrode solution (0.3% weight/volume).

Electrode resistances were between 4 and 7 MΩ, with access resistance values between 15 and 38 MΩ. Seal resistance values were always greater than 2 GΩ. Capacitance was fully compensated in voltage clamp during the on-cell configuration prior to breaking into the cell. For current-clamp recordings, full bridge balance compensation was used. Series resistance compensation between 45–65% was used during voltage clamp recordings. Voltage trace signals were amplified and low-pass filtered at 10–20 kHz before being digitized at 20–50 kHz. For current traces, signals were low pass filtered at 4 kHz. All electrophysiology was carried out using a Multiclamp 700B (Molecular Devices) and a Digidata 1550 (Molecular Devices). Liquid junction potentials were not corrected.

## Data analysis

Methods were as in [15], with all data taken from recordings collected in that study, and summarized above. Resting membrane potential was obtained by averaging across 1 s of the recorded membrane potential in the absence of an external input. Input resistance was calculated using the inverse of the slope of the linear fit to the steady state current voltage (I-V) relationship measured between 0 and 100 pA of injected current in 25 pA increments. The membrane time constant was obtained by fitting the voltage trace to a single exponential during the return to RMP after a 100 pA hyperpolarizing current step. For frequency-current measures, a series of depolarizing current steps of 25 pA were used to depolarize neurons and drive action potential generation. Some of the recorded neurons generated a weak early spike frequency adaptation, which was replicated in the model by accumulation of Na inactivation and Kv1 activation within a few successive spikes (Fig 1A2). The f/I curves shown in Fig 1B show the steady-state frequency after a weak early spike frequency adaptation. Only values at

which repetitive firing could be sustained were plotted. Recorded neurons also presented a much weaker and slower later adaptation, likely due to A-type potassium currents [89], which we omitted from the model for simplicity. For gap junction measures, a square hyperpolarizing pulse between -100 and -300 pA was used to hyperpolarize the pre-synaptic cell across 25–50 trials that were averaged in the post-synaptic cell. A measured junction potential of ~11 mV was not subtracted from recordings.

## Computational methods

All simulations were carried out in the BRIAN simulator [90]. The simulation code has been uploaded to modelDB at https://senselab.med.yale.edu/modeldb/enterCode?model=267338#tabs-1. The network consists of 100 single compartment model neurons with five state variables: the membrane potential, $V$, and four gating variables (m, h, n, and a) that use the same kinetic equations as the original Hodgkin-Huxley model [91,92], but with different parameters tuned to replicate the dynamics of fast spiking neurons in the mEC. Also, consistent with other models of fast-spiking interneurons [56,93], we included two delayed rectifier K$^+$ currents ($I_{Kv1}$ and $I_{Kv3}$). The differential equation for the membrane potential (V) of each neuron is $C_M dV/dt = I_{app} - I_{Na} - I_{Kv1} - I_{Kv3} - I_L - I_{syn} - I_{gap} - I_{ChR}$, where $C_M$ is the membrane capacitance, $I_{app}$ is an externally applied current that is only nonzero when simulating step currents for electrophysiological measurements, $I_{Na}$ is the fast sodium current, $I_L$ is the passive leak current, $I_{syn}$ is the GABA$_A$ synaptic current, $I_{gap}$ is the gap junctional current and $I_{ChR}$ is the simulated sinusoidal optogenetic drive. The equations for the intrinsic ionic currents are as follows: $I_{Na} = g_{Na} m^3 h(E_{Na} - V)$, $I_{Kv1} = g_{Kv1} a^4(E_K - V)$, $I_{Kv3} = g_{Kv3} n^4(E_K - V)$ and $I_L = g_L(E_K - V)$, with $E_{Na}$ = 50 mV, $E_K$ = -90 mV and $E_L$ varied across the population. The dynamics of the gating variables are given by $dx/dt = \alpha_x(1 - x) - \beta_x x$ for the activation variables (m, n, a) and by $dx/dt = \beta_x(1 - x) - \alpha_x x$ for the inactivation variable h, where $\alpha_x = k_{1x}(\theta_x - V)/(\exp((\theta_x - V)/\sigma_{1x}) - 1)$ and $\beta_x = k_{2x}\exp(V/\sigma_{2x})$ using parameters in Table 1.

In order to calibrate the network as described in the text accompanying Fig 1, we generated 100 vectors of length three, each with the three passive parameters: membrane time constant, input resistance and leakage reversal potential. The values for these parameters were drawn from random variables with uniform distribution in ranges similar to the ones observed in the recorded neurons, i.e. time constant from 3 to 7 ms, input resistance from 50 to 150 MOhm, and leakage reversal potential from -80 to -60 mV. The time constant and input resistance were used to set the membrane capacitance: $C_m = \tau_m/R_{input}$. Some of the parameters (see Table 1) for the voltage-gated currents were uniform across the 100 simulated neurons, but seven active parameters were varied. We generated another set of 100 vectors of length 7 from uniform distributions for the peak conductances for the three voltage-gated ion channels, Nav ($g_{Na}$ 6000 to 35000 nS), Kv1 ($g_{Kv1}$ 15 to150 nS) and Kv3/Na$_V$ ratio ($g_{Kv3}/g_{Na}$ 0.03 to 0.05) and the activation mid-potentials for the four gating variables, activation and inactivation of Nav ($\theta_m$ -60 to -45 mV, $\theta_h$ -60 to -50 mV), and activation of Kv1 ($\theta_a$ 35 to 55 mV), and Kv3 ($\theta_n$ -15 to 25 mV). Constraining the Kv3/Na$_V$ ratio helped reproduce some of the features from the experimentally recorded traces. Combining each of the 100 passive sets with each of the active sets produced 10,000 parameter sets, from which we selected 100 neurons as described in the text for Fig 1. The simulated f/I curves in Fig 1A2 and 1B2 and the simulations in Figs 3, 6A1 and 6B1 did not include gap junctions. In simulations without gap junctions, the leakage reversal potential $E_L$ was equal to the resting membrane potential.

The connection probabilities and conductances for both electrical and chemical synapses were taken from [15]. Although a single PV+ basket cell in region CA1 makes chemical synaptic contacts with about 60 other PV+ basket cells [79], we are unaware of similar data in the

mEC. For this reason, this number is not well-constrained in our model. In the mEC, the probability of both electrical and chemical synapses drops off dramatically at distances between somata that are 125–150 μm apart [15]. In order to apply the connection probabilities, the average number of PV+ cells that are within that distance of a typical soma of a PV+ cell must be estimated. This estimate can be refined by the size of the area illuminated by the laser. The diameter of this area is about 200 μm [12]. Thus, we conservatively estimated that each cell makes 36 chemical synaptic contacts onto other PV cells activated by the ChR2, with a maximum of 27 electrical contacts, which aligns with the measured connectivity probabilities if we assume a network of 100 neurons.

Chemical synapses were modeled by an inhibitory postsynaptic conductance with a biexponential waveform $g_{i,k}(t) = F(\exp(-(t - t_i^k - \delta_i)/\tau_1) - \exp(-(t - t_i^k - \delta_i)/\tau_2))$ where F is a normalization factor that sets the peak to one [94]. This conductance waveform was initiated after a conduction delay, δ by each spike $k$ in the presynaptic neuron $i$:
$I_{syn} = \sum_i \sum_k g_{syn,i} g_{i,k}(t)(E_{syn} - V)$. No spatial structure was assigned to the network, and the conduction delay for each synapse was assigned from a uniform distribution ranging from 0.6 to 1.0 ms. The delays, a fixed $\tau_2$ of 0.3 ms (which corresponds to a rise time of 0.34 ms, see [94]), and a fixed decay time constant $\tau_1$ of 2.0 ms, were calibrated according to the experimental data in [15]. The reversal potential $E_{syn}$ was varied to simulate shunting (-55 mV) and hyperpolarizing inhibition (-75 mV). The maximal synaptic conductance was lognormally distributed [95] with parameters μ = 0 corresponding to the log of 1 nS and σ = 1. The probability of connection in each direction was 0.36.

The gap junction current is given by $I_{gap} = \sum_i g_{gap,i}(V - V_i)$ summed over the $i$ neurons connected to a given neuron. When gap junctions are included, the measured input resistance is no longer determined by the leakage conductance alone, but also by the gap junctional conductance. To first order, one can approximate the input resistance for each neuron $i$ by $R_{input,i} = 1/(g_L + \sum_i g_{gap,i})$. We used a 27% probability that any pair of neurons was connected by gap junctions [32]. Peak conductances were obtained from data originally collected for [32]. The histogram of peak conductances for the electrical synapses in Fig 2B suggests a bimodal distribution with a weak mode given by the positive half of a Gaussian distribution with zero mean and a standard deviation of 0.4 nS, and a strong mode, which we approximated by a Dirac delta at 1.2 nS. For each neuron $i$ in the network, we chose 27 other neurons at random, labeled $j$. We then assigned a 25% probability that a particular connection was of the strong mode type, and 75% that it was of the weak mode type. For strong mode synapses, a value of 1.2 nS was assigned to $g_{ji}^{gap}$ whereas for the weak mode it was drawn from the truncated Gaussian with s.t.d. 0.4 nS. Each gap junction was added bidirectionally with the same strength, and $g_{ji}^{gap}$ was subtracted from the leakage conductance of both neurons to keep $R_{input}$ in the experimentally constrained range. Gap junctions were only added while the leakage conductance remained above a predefined floor value $g_L^{min}$ (1.5 nS), since this quantity cannot be zero or negative. Frequently the total number of electrical synapses was less than 27 for a given neuron, but this procedure was necessary to honor the data since passive properties were measured with gap junctions intact. The network synchrony and dominant frequency were not very sensitive to the $g_L^{min}$ parameter for values between 1 and 2 nS (S5 Fig). The capacitance was not changed because the input resistance was approximately preserved. Leakage reversal potentials were then adjusted to preserve the experimentally constrained distribution of resting membrane potentials.

The optogenetic drive is present in the network simulations as
$I_{ChR} = g_{ChR}\sin(2\pi f t/1000)(E_{ChR} - V)$, where t is in ms, f is 8 Hz, $E_{ChR}$ is 0 mV, and $g_{ChR}$ is 14 nS, unless otherwise specified.

Synaptic depression was calibrated according to [32] using the model by Markram and Tso-dyks [96] adapted by [97], but neglecting facilitation. In this model, the available fraction of transmitter x evolves according to: $\frac{dx}{dt} = \frac{1-x}{\tau_r} - U_{SE}x\delta(t - t_k)$, where $t_k$ is the $k$th spike time, $\tau_r =$ 100 ms is the recovery time to replenish the available pool of vesicles for release, $U_{SE} = 0.3$ is the fraction of available pool released by each spike, and the value of x prior to a spike is pro-portional to the peak current value of the inhibitory post-synaptic current.

A forward Euler method was used to integrate the ODEs. For the calibration of passive and active properties, and for the network simulations without gap junctions, an integration time step of 0.01 ms was used. Simulations did not converge with that time step when heteroge-neous gap junction conductances were considered, and the time step was reduced to 5.0e-4 ms. A 4th order Runge-Kutta integration method produced identical results. The histograms in Figs 3–6, 8 and 9 were computed using a bin width of 0.1 ms.

The frequency, power and theta phase onset and offset for the fast oscillations were obtained from Continuous Wavelet Transform powers like the example shown in Fig 8C. The wavelet transforms were computed using the *cwt* function from *scipy.signal* with a Morlet wavelet (*Morlet2* in *scipy.signal*) applied to the population rate. In particular, the wavelet width parameter was $\omega_0{}^*f_s/(2^*\pi^*f)$ where $\omega_0 = 5$ is the wavelet order, $f_s$ is the sampling rate and $f$ is the frequency. The wavelet power was obtained as the squared modulus of the complex wavelet transform. The population rate used to compute it was obtained using a flat sliding window of width 0.1 ms (*sampling_rate*) in the function *PopulationRateMonitor.smooth_rate()* from BRIAN 2. The considered range of frequencies were from $f = 50$ to 449 Hz in steps of 3 Hz. No normalization was applied.

The dominant frequency, $f_{max}$, or frequency of maximum power in Fig 7, was computed from all theta cycles for each connectivity pattern as the frequency with maximum wavelet power. The mean and standard deviation of the dominant frequency were computed only from cycles whose power was above 0.3 times the maximum power recorded for that connec-tivity pattern in order to avoid contamination by cycles in which no synchrony was observed.

The circular histograms for onset and offset theta phases of the fast oscillations were com-puted using the *rose.diag* function in the *circular* package from CRAN R by pooling together the 900 values corresponding to each parameter set, i.e. the 30 last simulated theta cycles for each of the 30 networks with different connectivity pattern. The first four cycles were dis-carded to eliminate transients.

## Supporting information

**S1 Fig. Histograms of passive and active properties of experimental (left) and model neu-rons (right).** A. Time constants. B Input resistance. C Resting Potential. D. Cutoff frequency E. Rheobase.
(TIF)

**S2 Fig. f/I curves for model network in Fig 6A2 and 6B2 with gap junctions intact. A.** Curves for one network instantiation. Adding gap junctions at a few values of injected current on a small number of model neuron f/I curves, possibly due to rebound spiking in another neuron strongly coupled to the injected neuron. B. An example from a pair of PV cells con-nected by gap junctions showing the gap junction current recorded under voltage clamp at -40 mV in one neuron during a single action potential generated with a brief, large pulse of current (dashed lines) in the other neuron. The bulk of the gap junction current flowed during the downstroke of the spike and the subsequent AHP [42].
(TIF)

**S3 Fig. Simulations in Fig 6 rerun with conduction delays set uniformly to 1.6 ms produce similar results.**
(TIF)

**S4 Fig. Effect of increasing chemical synapse strength in shunting networks with unphysiologically strong gap junctions.**
(TIF)

**S5 Fig. Values for $g_L$ floor do not substantially affect oscillatory frequency or power.**
(TIF)

## Author Contributions

**Conceptualization:** John A. White, Carmen C. Canavier.

**Data curation:** Guillem Via.

**Formal analysis:** Guillem Via.

**Funding acquisition:** John A. White, Carmen C. Canavier.

**Investigation:** Guillem Via, Roman Baravalle, Fernando R. Fernandez, Carmen C. Canavier.

**Project administration:** Carmen C. Canavier.

**Resources:** John A. White.

**Software:** Guillem Via, Roman Baravalle.

**Supervision:** John A. White, Carmen C. Canavier.

**Validation:** Roman Baravalle.

**Visualization:** Guillem Via, Roman Baravalle, Fernando R. Fernandez, Carmen C. Canavier.

**Writing – original draft:** Guillem Via, Fernando R. Fernandez, Carmen C. Canavier.

**Writing – review & editing:** Guillem Via, Roman Baravalle, Fernando R. Fernandez, John A. White, Carmen C. Canavier.

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
