## [Decision Letter · Decision Letter 0]

20 Jun 2022

Dear Dr. Canavier,

Thank you very much for submitting your manuscript "Interneuronal network model of theta-nested fast oscillations predicts differential effects of heterogeneity, gap junctions and short term depression for hyperpolarizing versus shunting inhibition" for consideration at PLOS Computational Biology.

As with all papers reviewed by the journal, your manuscript was reviewed by members of the editorial board and by several independent reviewers. In light of the reviews (below this email), we would like to invite the resubmission of a significantly-revised version that takes into account the reviewers' comments.  We note that the reviewers all found the work to be interesting and that the main comments mostly relate to clarity of exposition of a rather complex collection of fundings, clarification of some specific points, and ensuring that the paper is properly placed in the context of existing literature, with appropriate citations and description of what is novel in this work.

We cannot make any decision about publication until we have seen the revised manuscript and your response to the reviewers' comments. Your revised manuscript is also likely to be sent to reviewers for further evaluation.

Sincerely,

Jonathan Rubin

Associate Editor

PLOS Computational Biology

Kim Blackwell

Deputy Editor

PLOS Computational Biology

Reviewer's Responses to Questions

**Comments to the Authors:**

Reviewer #1: In this paper, the authors organize an inhibitory model with the goal to dissect the role of multiple factors on network activity. The factors considered are 1) nature of inhibitory coupling (shunting vs hyperpolarizing), 2) heterogeneity in intrinsic or synaptic parameters, 3) presence/absence of gap junctions and 4) presence/removal of short-term depression. They frame the analysis as a potential duality in the network, which is either coupled with hyperpolarizing or shunting connections. They take steps to ensure that the starting network they work with is representative of biophysical properties of mEC, as they are mindful of differential characteristics of similar cell types across regions.

Among the many points that the manuscript makes, a combination of all the factors considered is necessary for the carefully tuned model to reproduce dynamics found in experimental results.

The extensive and comprehensive analysis allows readers to navigate a quite complex picture, which is explored in depth. I find the analysis methodologies and the logic of the paper very solid. I have enjoyed learning about this model, the authors perspective and the way the paper has been organized to help the reader access the progressively more complex picture.

I have very few points that I would suggest could further improve this work.

1. Despite the great articulation of the topic that authors provide, the paper by its own nature has many compounding parts. As a result, it is hard to leave this paper with a unified picture of what one learns from this work. To possibly mitigate this factor, I would like to suggest that, in the spirit of delivering a comprehensive yet succinct message that the reader can ‘take home’, findings could be summarized in some schematic representation that recapitulates the main roles that each factor plays in influencing network dynamics. I am referring to a pictorial representation, not actual simulations or data, since those are appropriately proposed in the collection of figures shared with the reader.

2. Line 179 (Figure 4). The authors state: “the perfect synchrony present with chemical synapses intact indicates that the synapses themselves are not sufficient to destabilize global synchrony in the presence of a common sinusoidal drive”. Does that imply that you have checked and the synchronous state you see with synapses present is the same as the synchronous state you will get without any synapse? Or, while one finds global coordination in both cases, the synapses do introduce some difference in the rhythm? I am thinking of mean fields, and how to assess the ‘field response’ to input one would consider both the intrinsic properties of the cell population (the f-I curves, for example) and the synaptic currents within the network to establish the regional response to input. So, I am guessing that synapses still do something, possibly even something that promotes stability, rather than de-stabilize? I feel this question is partly answered later in the paper, but for a reader it arises at this point, so it should be at least partially addressed here.

3. Regarding a step in the methodology to fit model parameters to experimental properties. What is the rationale for choosing uniform sampling (as opposed to other types, say Gaussian) in a network of 100 neurons? Were the values found among the real cells uniformly distributed?

4. Line 186: “However, the peaks widen and diminish in height until fast oscillations disappear. The synchronization properties of the same network using shunting chemical synapses also decays but not as abruptly.” Did you look at these time scales? Is this effectively a transient gamma inside some theta phases? Again, I feel this is partly addressed later, but the reader encounters the question at this stage. Perhaps just a brief sentence to acquiesce curiosity at this stage is all that’s needed.

5. Figure 5. I enjoyed the result section pertinent to figure 5 and appreciate the effort put in delivering intuitive understanding of PRC and their usefulness to a broader audience. However, I would encourage the authors to consider splitting figure 5A in two figures, one for the shunting network and one for the hyperpolarizing current network. This is because it would strongly simplify following along the text from the figure, which currently has too many facts crowded together. As a result, one ‘sees’ all these facts as once when referred to the figure, but only gets to understand what all those curves and arrows and ranges are doing much later in the text.

6. A few small things:

A: Typo line 518 pars “reticulate” should be pars reticulata

B: Typo in line 105, a comma inside the parenthesis

C: You mean figure 2C1 and 2C3 (not B) in line 143 (I think)

Reviewer #2: See the attached file in pdf with my remarks/comments

Reviewer #3: The ms. by Via et al addresses the mechanisms of high-frequency network oscillations in the medial entorhinal cortex (mEC) using computational analysis. It focuses on inhibition based oscillation model (ING) in an cell type-specific (fast-spiking parvalbumin-positive basket cells) optogenetic theta-patterned activation paradigm. Intrinisic properties of the modelled interneurons and their synaptic connections (chemical and electric) are based on experimental data (partially published in a previous paper Ref 32); the authors made particular attention to the heterogeneity in the parameters among the interneurons.

The main finding of the study is that hyperpolarizing inhibition supports the generation of high-frequency oscillations more reliably than shunting inhibition when experimentally determined properties (incl. type 2 excitability) with realisitc heterogeneities of the cells were implemented in the model. Fast oscillations emerge on the upstroke of the theta wave if short-term plasticity of the inhibitory synapses was also implemented. Finally, gap junctions support the oscillations and are required for their generation if inhibition is shunting.

The results are compelling and support the conclusions. The insights gained are important and help to understand how fast-oscillations are generated in the entorhinal cortex. In view of their high functional relevance in learning and memory, and spatial navigation, the study would be of interest to a broader neuroscience audience.

The results, in general, are well presented, however, some details, that are important to the reader for a full understanding, are missing or not fully explained:

- The model considers only the interconnected interneuron network, but excludes principal cells. This is a valid approach to understand the mutually interconnected "pacemaker" interneuron network, but raises the question if interneurons cna entrain the activity of the pyramidal cells at the same high frequencies, given the differences in the IN-IN and IN-PC synaptic connections.

- Basic information on the electrophysiological recordings is completely missing from the ms. Only a reference is made to the previous publication from which data was partially taken. Please provide relevant information in the Methods, including how many new recordings were made for this study and how neurons/ from the previous study were selected for this one.

As the aim here was to calibrate heterogeneity, it is surprising that only 11 interneurons were evaluated in contrast to the 122 pairs recorded on Ref 32. If data was taken form the previous publication, why wasn't the whole data taken to get the best possible estimate of heterogeneity?

Finally, it would be also helpful to the reader if the most important aspects (e.g. whole-cell recording in slice preparation, species, mEC layer) would be included in one or two sentences.

- The frequency range of the oscillations investigated remains poorly defined. While the Abstract and Introduction focus on gamma oscillations, in the Discussion only ripple oscillations are mentioned. While this has no relevace for the resutls and conclusions, an inconsistent presentation will confuse the readers.

An indirectly related issue is the value of the membrane time constant - the mean value is 5 ms (range 3 - 7 ms), which is approximately twice as fast than other estimates for hippocampal, but also EC basket cells (~10 ms) and also almost 2 ms faster than the mean value in Ref 32 (6.8 ms). This difference might be due to the large current (100 pA) applied to produce the voltage responses for the calculations - with these large currents it is likely that voltage-gated conductances are also activated.

What would be the effect of a slower time constant on the results and conclusions?

- Synaptic delays were set between 0.6 -1.0 ms, but it is not clear from the text if there was any spatial rule (i.e. conduction times) or wether they were set randomly. The range corresponds well to a axonal arbor with 150 um radius in L2 of the mEC.

In fact, the short range of connectivity of basket cells is a feature of the mEC which contrasts the wide axonal arbors of basket cells (500 - 1000 um) in the hippocampus/dentate gyrus. Please consider this difference and its impact between the models for the hippocampus and the mEC in the discussion.

- In Figure 4 legend A2 & 3 and B2 & 3 are not explained.

Text repeatedly refers to non-existing Figure 4C (e.g. line 210, 286, 319 and Fig 5 legend) .

- I find it somewhat inconvenient that the panel labels in the figures are not aligned to the left of the panels, for easy orientation, but have variable positions and are merged with the titels (in some without any space between them, e.g. Figure 5BCD).

- Thre are some minor typos and punctuation errors (e.g. lines 103, 105, 219).

**Have the authors made all data and (if applicable) computational code underlying the findings in their manuscript fully available?**

Reviewer #1: Yes

Reviewer #2: **No: **

Reviewer #3: Yes

PLOS authors have the option to publish the peer review history of their article (what does this mean?). If published, this will include your full peer review and any attached files.

Reviewer #1: No

Reviewer #2: No

Reviewer #3: **Yes: **Imre Vida
---

## [Decision Letter · Decision Letter 1]

14 Nov 2022

Dear Dr. Canavier,

We are pleased to inform you that your manuscript 'Interneuronal network model of theta-nested fast oscillations predicts differential effects of heterogeneity, gap junctions and short term depression for hyperpolarizing versus shunting inhibition' has been provisionally accepted for publication in PLOS Computational Biology.

Best regards,

Jonathan Rubin

Academic Editor

PLOS Computational Biology

Kim Blackwell

Section Editor

PLOS Computational Biology

Reviewer's Responses to Questions

**Comments to the Authors:**

Reviewer #1: the authors addressed all my points satisfactorily.

Reviewer #2: The authors answered to all my questions in a satisfactory way and modified accordingly the text. Therefore I recommend the publication of the amended manuscript in PLOS Computational Biology.

**Have the authors made all data and (if applicable) computational code underlying the findings in their manuscript fully available?**

Reviewer #1: Yes

Reviewer #2: Yes

PLOS authors have the option to publish the peer review history of their article (what does this mean?). If published, this will include your full peer review and any attached files.

Reviewer #1: No

Reviewer #2: No

---

## [Editor Report · Acceptance letter]

28 Nov 2022

PCOMPBIOL-D-22-00574R1 

Interneuronal network model of theta-nested fast oscillations predicts differential effects of heterogeneity, gap junctions and short term depression for hyperpolarizing versus shunting inhibition

Dear Dr Canavier,

I am pleased to inform you that your manuscript has been formally accepted for publication in PLOS Computational Biology. Your manuscript is now with our production department and you will be notified of the publication date in due course.

With kind regards,

Olena Szabo
